# snPATHO-seq, a versatile FFPE single-nucleus RNA sequencing method to unlock pathology archives
Taopeng Wang [1,2,26], Michael J. Roach [3,4,5,26], Kate Harvey [1], Javier Escudero Morlanes[6], Beata Kiedik [1,2], Ghamdan Al-Eryani[1,2], Alissa Greenwald [7], Nikolaos Kalavros [8,9,10,11], Felipe Segato Dezem[12], Yuling Ma[8,9,10,11], Yered H. Pita-Juarez[8,9,10], Kellie Wise[3,4], Cyril Degletagne[13], Anna Elz[14], Azi Hadadianpour[14], Jack Johanneson [14], Fiona Pakiam[14], Heeju Ryu[15], Evan W. Newell [14,15], Laurie Tonon [13,16], Andrew Kohlway[17], Tingsheng Drennon [17], Jawad Abousoud[17], Ryan Stott[17], Paul Lund [17], Jens Durruthy[17], Andres F. Vallejo [18], Wenyan Li[19,20], Robert Salomon [19,20], Dominik Kaczorowski [21], Joanna Warren[21], Lisa M. Butler [4,22], Sandra O'Toole[1,2,23,24,25], Jasmine Plummer [12], Ioannis S. Vlachos [8,9,10,11], Joakim Lundeberg [6], Alexander Swarbrick [1,2,27] ✉ & Luciano G. Martelotto [3,4,27] ✉

Formalin-fixed paraffin-embedded (FFPE) samples are valuable but underutilized in single-cell omics research due to their low RNA quality. In this study, leveraging a recent advance in single-cell genomic technology, we introduce snPATHO-seq, a versatile method to derive high-quality single-nucleus transcriptomic data from FFPE samples. We benchmarked the performance of the snPATHO-seq workflow against existing 10x 3' and Flex assays designed for frozen or fresh samples and highlighted the consistency in snRNA-seq data produced by all workflows. The snPATHO-seq workflow also demonstrated high robustness when tested across a wide range of healthy and diseased FFPE tissue samples. When combined with FFPE spatial transcriptomic technologies such as FFPE Visium, the snPATHO-seq provides a multi-modal sampling approach for FFPE samples, allowing more comprehensive transcriptomic characterization.

Clinical tissue samples are routinely preserved as formalin-fixed paraffin-embedded (FFPE) samples. Most pathology laboratories in the US process between 10,000 and 100,000 FFPE blocks annually, representing a significant resource for clinical research[1]. Advances in molecular technologies such as genomic, transcriptomic, and proteomic techniques have expanded the use of FFPE samples for molecular characterization[2–5]. However, these applications are often conducted at the bulk level and cannot fully resolve the molecular heterogeneity in cancers and other biological settings. Considering the abundance of clinical FFPE samples, technologies enabling single-cell profiling of FFPE samples hold immense potential to advance human health research.

Previously, our team successfully conducted single-nucleus genomic profiling using clinical human FFPE tissue samples, demonstrating the feasibility of isolating intact nuclei from FFPE samples for molecular characterization[6]. However, extending this technology to transcriptomics poses a significant challenge due to RNA fragmentation in FFPE samples caused by formalin fixation, high heat, and paraffin embedding[2]. Many conventional single-cell RNA sequencing (scRNA-seq) technologies, such as 10x 3' and SMART-seq, rely on the generation of cDNA libraries through poly(dT) probe capture and reverse transcription of intact mRNA molecules[7,8]. This dependency on RNA integrity makes these protocols suboptimal for FFPE samples.

An emerging strategy for gene expression profiling in FFPE samples is RNA-binding probes. This strategy has been widely adopted in spatial transcriptomic technologies, such as MERFISH, 10x FFPE Visium, and 10x Xenium[9–11]. Probes used in these assays target short sections (e.g., 50 bp in 10x FFPE Visium assay) of the RNA molecules, making this gene expression detection strategy more resilient against RNA fragmentation. However, these spatial transcriptomic technologies cannot provide transcriptomic-level molecular characterization (e.g., MERFISH and 10x Xenium) or single-cell resolution (e.g., 10x Visium) due to technical limitations. Therefore, scRNA-seq remains a powerful tool for unbiased molecular characterization of tissue samples, which can enhance the interpretation of spatial transcriptomics data when integrated.

In 2022, 10x Genomics released a new chemistry, the 10x Flex assay, adopting a similar probe design from the 10x FFPE Visium assay to scRNA-seq[12]. The default workflow is designed for fresh or frozen samples, where the samples are first fixed and prepared into a single-cell/nucleus suspension before hybridizing with the Flex probes. Upon capturing the cells/nuclei, the probes, instead of the RNA transcripts, are barcoded and processed into a sequencing library for gene expression profiling. Since only small sections of the RNA molecules are targeted, the 10x Flex chemistry is suitable for formalin-fixed samples where RNA integrity is impaired.

In light of this technical advancement, we have developed a nuclei isolation protocol tailored for FFPE samples. Our protocol includes serial steps of rehydration, enzyme-based tissue dissociation, and nuclei isolation prior to gene expression analysis using the 10x Flex assay. The combined nuclei isolation and gene expression analysis is termed snPATHO-seq. We tested the snPATHO-seq workflow on three human breast cancer tissue samples and demonstrated its concordance with the conventional 10x 3' and 10x Flex chemistries that had only been released for fresh or frozen samples at the time. During manuscript preparation, 10x Genomics released a demonstrated protocol to enable scRNA-seq profiling of FFPE samples using the same Flex chemistry[13]. A major difference between the snPATHO-seq and the scFFPE workflows is the lack of cell membrane lysis and nuclei isolation processes in the scFFPE workflow. We directly compared the scFFPE to the snPATHO-seq and observed the isolation of both intact nuclei and cells from the FFPE tissue samples using the scFFPE protocol. More recently, 10x Genomics has amended the scFFPE protocol to highlight its nature in isolating nuclei instead of cells from FFPE samples despite the lack of a nuclei isolation procedure[14]. While the scFFPE protocol can detect single-cell transcriptomic signatures representative of the biology in many FFPE tissue samples tested, we demonstrated higher robustness of the snPATHO-seq on the tissue samples tested, establishing it as a more mature protocol for FFPE single-nucleus study. We release the snPATHO-seq protocol and the data used for the evaluation to interested researchers aiming to foster growth in the scientific community focused on FFPE single-cell research.

## Results and discussion
### 10x Flex chemistry offers robust cell-type signature detection with reduced transcriptomic coverage

Given that the 10x Flex chemistry is a relatively new assay with limited performance evaluations, we first compared it to the well-established 10x 3' assay using two replicates of PBMC samples from the same donor (Fig. 1a, Supplementary Data 1). When standardized to the same sequencing depth (~30,000 reads per cell), both the Flex and 3' assays detected comparable numbers of unique molecular identifiers (UMIs) and genes (Fig. 1b, c). Principal component analysis (PCA) revealed a clear separation of the collected data by cell lineage (i.e., T cells (*CD3D*), B cells (*MS4A1*), and myeloid cells (*CD14*)) in the PBMC samples (Supplementary Fig. 1a). In addition, we detected a separation of the Flex and 3' data along PC2 (Supplementary Fig. 1b, c). Further investigation highlighted several MHC class II genes among the top drivers of PC2 (Supplementary Fig. 1d). While these genes are critical functional markers for immune cells, they were excluded from the Flex probe panel due to high inherent allelic diversity[15]. Thus, caution is advised when selecting the Flex assay for scRNA-seq gene expression readouts.

Nonetheless, we were able to identify comparable cell populations in Flex and 3' data following Seurat CCA integration (Fig. 1d). The Uniform Manifold Approximation and Projection (UMAP) analysis showed no obvious deviation between the data collected using the two methods (Fig. 1e). We validated the detected cell type signatures through three different methods: expression of canonical cell type markers (Fig. 1g), expression of the top 200 differentially expressed genes (Supplementary Fig. 2a), and annotation of a public dataset[16] using cell type signatures from the current study (Supplementary Fig. 2b, c). We obtained good concordance between the Flex and 3' data across all three analysis methods tested, suggesting the Flex assay can detect comparable cell type signatures as the 3' assay.

Interestingly, while there was no assay-specific cell type distribution, innate lymphoid cells (ILCs) were more abundant in the Flex data than in the 3' data across both technical replicates (Fig. 1f). ILCs are challenging to detect with conventional scRNA-seq methods due to the low RNA content[17,18]. Considering that the Flex probes target small fragments of the RNA molecules with a reduced requirement for RNA quality compared to the conventional 3' assay, the increased ILC proportion likely reflects the advantage of the Flex assay in detecting cell populations with low or degraded RNA content.

### The snPATHO-seq workflow combines nuclei isolation with 10x Flex chemistry to enable snRNA-seq profiling on FFPE samples

The observations above gave us confidence in the performance of the 10x Flex assay as a substitute for the 10x 3' assay with decreased transcriptomic coverage. At the time of the first release, the Flex assay was only compatible with fresh or frozen samples. However, considering the similar design of the Flex probes to the 10x FFPE Visium probes, we reasoned that the Flex assay could be adapted for FFPE samples. We then designed a novel snRNA-seq workflow, snPATHO-seq, tailored for human clinical FFPE tissue samples based on the Flex assay (Fig. 2a). Our workflow features the extraction of intact nuclei from archival FFPE samples, which are then used for transcriptomic profiling using the 10x Flex assay. We were able to generate snRNA-seq data from 15 FFPE samples tested using the snPATHO-seq workflow (Supplementary Data 1). In addition, the snPATHO-seq protocol has been adopted in recent studies[19,20], and a more in-depth workflow illustration has been released online[21,22].

We tested the performance of the snPATHO-seq on three human breast cancer samples, including one primary HER2-amplified breast cancer sample (4066) and two liver metastases (4399: TNBC and 4411: Luminal) (Supplementary Data 1). Phase-contrast microscopy confirmed the isolation of intact nuclei from the FFPE samples tested (Supplementary Fig. 3a). Given that no other FFPE snRNA-seq workflow was available at the time, we compared the performance of the snPATHO-seq to the conventional 10x 3' (frozen-3') and Flex (frozen-Flex) assays using frozen tissues preserved from the same tumor lesions. We did observe a decrease in the number of UMIs and genes detected per nucleus in the snPATHO-seq data compared to the frozen-Flex and frozen-3' data, potentially due to the lower RNA quality of the FFPE samples (Fig. 2b, c). Nonetheless, the snPATHO-seq and the frozen-Flex data showed a good overlap in the lower dimensional UMAP space without data integration (Fig. 2d, e). Using the Seurat CCA method, we derived common cell type nomenclatures across all methods for each sample. For samples 4066 and 4411, the identified cell populations matched the morphological features of the tissue samples. Notably, we identified myoepithelial cells known to be associated with normal mammary glands and ductal carcinoma in situ (DCIS) in 4066 and liver resident cell types (e.g., hepatocytes, cholangiocytes and liver sinusoidal endothelial cells (LSECs)) in 4411 (Fig. 2f, Supplementary Figs. 4a and 5a, c). The identified cell type proportions were generally comparable between the snRNA-seq workflows tested for samples 4066 and 4411, while the concordance was better between the snPATHO-seq and the frozen-Flex data (Fig. 2h, Supplementary Fig. 4c). However, in sample 4399, the liver resident cell populations were underrepresented in the frozen-Flex and frozen-3' data (Supplementary Figs. 4f, g and 5b). While we could validate the presence of cancer-adjacent liver tissue in the FFPE sample, the frozen sample was exhausted after nuclei extraction and could not be used for morphology examination. Considering that the same cell populations could be identified by the frozen-Flex and frozen-3' workflows in sample 4411, the deviation observed in 4399 is likely due to sampling bias between the FFPE and the frozen samples.

In terms of the gene expression signatures, differential gene expression analysis revealed comparable expression patterns of top differentially expressed genes across the snRNA-seq workflows tested (Fig. 2i, Supplementary Fig. 4d, h). Cancer cell copy number inference analysis successfully predicted the amplification of the *ERBB2* gene in data from 4066, which is in

line with the clinical diagnosis of this tumor (Supplementary Fig. 6). In addition, the copy number profiles inferred using the snPATHO-seq, frozen-Flex, and frozen-3' data were generally comparable across the three breast cancer samples tested (Supplementary Fig. 6). Moreover, when mapped to the matching FFPE Visium data, we detected comparable spatial

cellular distribution patterns using cell type signatures detected by different snRNA-seq workflows from samples 4066 and 4411 (Supplementary Fig. 7a, c, d, e). Unsurprisingly, due to the absence of cell type signatures from liver resident cell populations, we observed substantial bias in spatial cellular enrichment pattern prediction using snRNA-seq data generated by

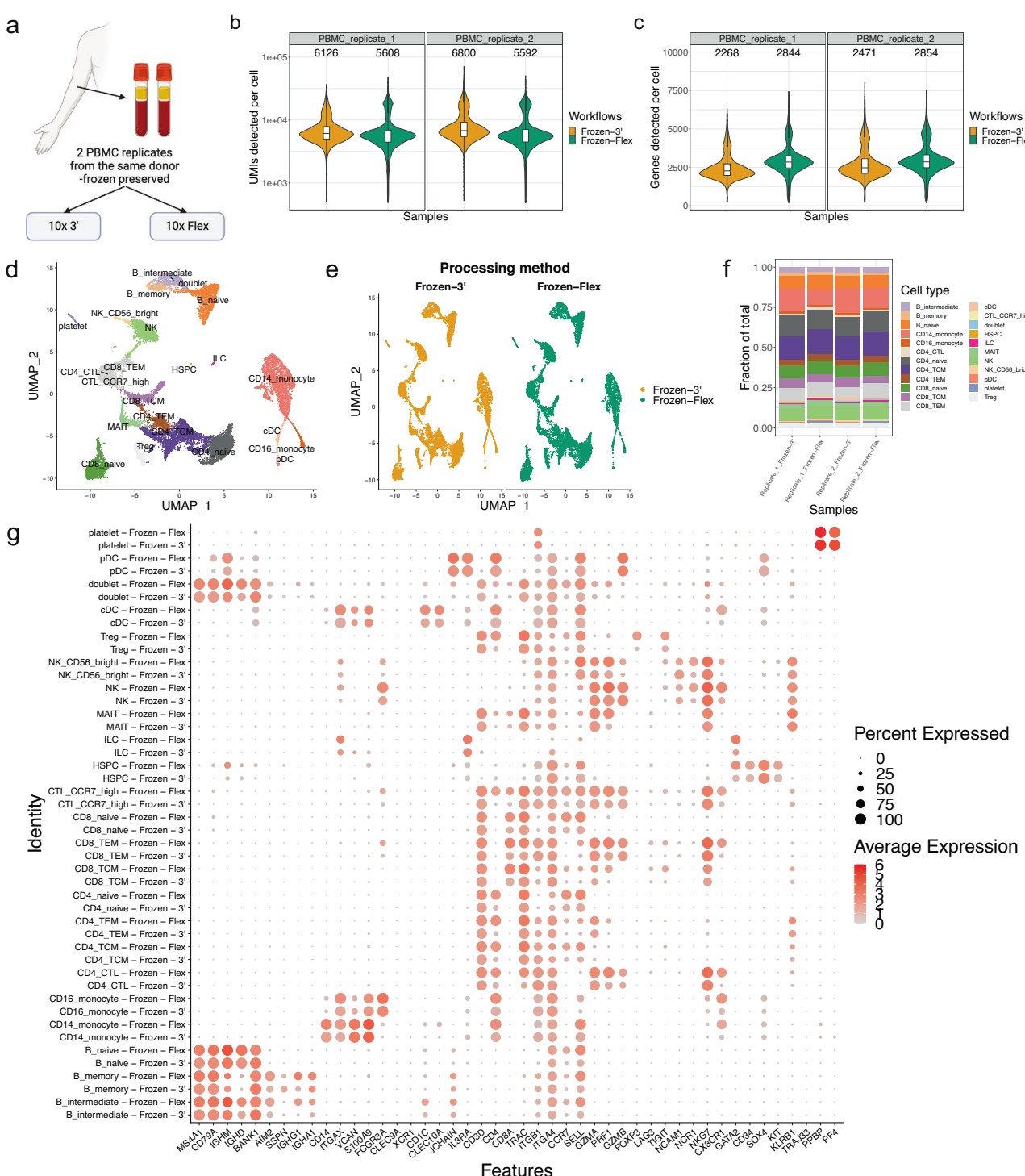

**Fig. 1 | The 10x single-cell Flex chemistry produces results comparable to those of 10x 3' chemistry when applied to PBMC samples. a** Illustration of experiment design for 10x 3' and Flex assay comparison. Created in BioRender. Wang, T. (2024) BioRender.com/l86j691. **b, c** Violin plots of the number of UMIs (**b**) and Genes (**c**) detected per cell. The boxes in the violin plots show the UMIs (**b**) and Genes (**c**) median and interquartile range. **d** UMAP embedding of PBMC cells integrated using the Seurat CCA method and annotated by cell type. **e** UMAP embedding split by processing method. **f** Barplots showing the fraction of cell types detected in each technical replicate processed using 10x 3' or Flex chemistry. **g** Dotplot of the expression of canonical PBMC cell type markers. *N* = 2 sample per protocol. The two replicates were conducted using samples from the same donor. Replicates are labeled in (**b, c, f**).

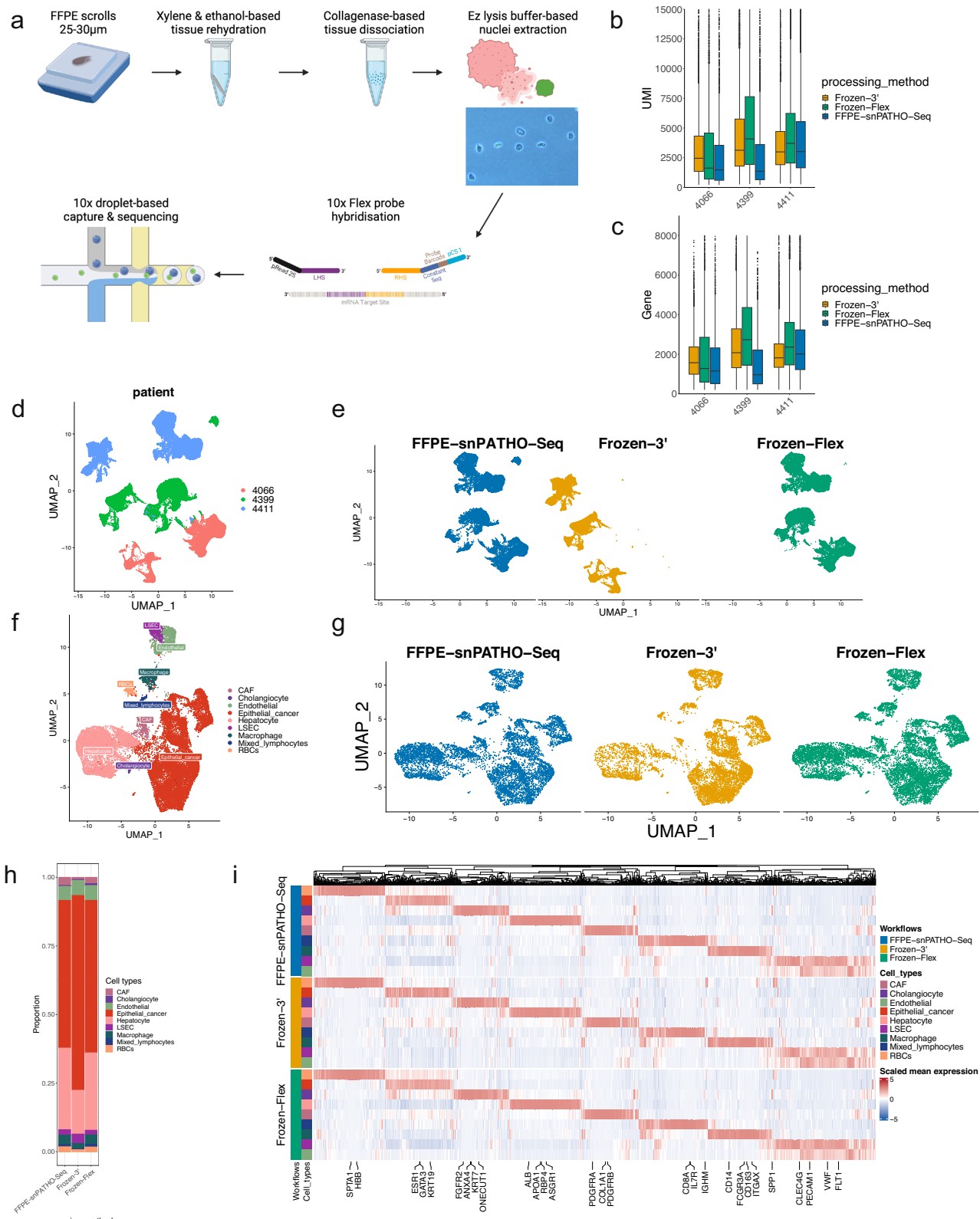

the frozen-Flex and the frozen-3' workflows from sample 4399 (Supplementary Fig. 7b, f). Together, these results demonstrated that the snPATHO-seq workflow enables accurate cell type identification using archival FFPE tissue samples and permits data integration with matching FFPE Visium data, allowing multi-modal sampling of the same FFPE sample.

## snPATHO-seq captures similar transcriptomic signatures in breast cancer cells as the existing snRNA-seq methods

In addition to the integration-based comparisons above, we also took a non-biased approach to compare transcriptomic signatures extracted from data generated using different snRNA-seq workflows[23]. This workflow operates on a per-dataset basis and allows us to extract 44 Non-negative matrix

**Fig. 2 | The snPATHO-seq workflow enables nuclei isolation and single-nucleus gene expression detection from human FFPE tissue samples. a** Illustration of the snPATHO-seq workflow. Created in BioRender. Wang, T. (2024) BioRender.com/u53s150. **b, c** Boxplots of the number of UMIs (**b**) and genes (**c**) detected per nucleus. The boxes show the UMIs (**b**) and Genes (**c**) median and interquartile range. Outliers were shown as dots. **d** UMAP embedding of unintegrated snRNA-seq data annotated by sample IDs. **e** UMAP embedding of unintegrated snRNA-seq data split by processing methods. **f** UMAP embedding of Seurat CCA integrated snRNA-seq data from patient 4411 annotated by cell type. **g** UMAP embedding of Seruat CCA integrated 4411 snRNA-seq data split by processing methods. **h** Barplot showing the fraction of cell types detected by different snRNA-seq methods in sample 4411. **i** Heatmap of the scaled expression of selected cell type markers detected by differential gene expression analyses in 4411 data. The top 200 significantly differentially expressed genes identified in each cell population (if available) were selected by fold change and used for plotting. A gene was considered significantly differentially expressed if the BH-adjusted *P* value was lower than 0.05. Genes were arranged by hierarchical clustering based on the expression in the FFPE-snPATHO-seq data on the x-axis. Cell types identified by different snRNA-seq workflows were manually arranged on the y-axis. *N* = 1 sample per protocol.

factorization (NMF) programs from each dataset (per sample, per snRNA-seq workflow). The derived NMF programs from all datasets were merged and filtered based on similarities, resulting in 57 robust NMF programs (Supplementary Data 2). Despite being derived from different samples and snRNA-seq workflows, the robust NMF programs shared many similar genes and could be further grouped into 15 clusters with biological significance (Fig. 3a, b, Supplementary Table 1, Supplementary Data 3). For example, C6 contained genes related to cell cycle progression (Supplementary Data 3). This signature was enriched in cancer cells marked by high *MKI67* gene expression and shared a similar enrichment pattern as a published cell cycle progression gene signature[23] (Fig. 3c i–iii). In addition, we identified a calcium signaling-related cluster from sample 4066 represented by genes including *ITPR2, CADPS2, CAPN13, NFATC4,* and *ADRGV1,* all pivotal to calcium regulation and signaling[24–28] (Supplementary Data 3). Moreover, mapping this signature to Visium data from 4066 revealed a spatial distribution pattern specific to DCIS, a condition often tied to calcification[29] (Fig. 3d).

Notably, while we identified many robust NMF programs derived from the same samples using different snRNA-seq workflows, the robust NMF programs derived from the snPATHO-seq and the Flex workflow generally showed more similarities to each other than to the robust NMF programs from 3' (Fig. 3a). For example, while the robust NMF programs related to C3 all contained genes *ITPR2* and *CAPN13,* the robust NMF program derived from the 3' workflow also contained several non-coding RNA genes (e.g., *AC091646.1, AC012501.2,* and *AC078923.1*) that were not included in the Flex probe panel (Supplementary Fig. 8a). We observed a similar result when comparing the gene composition of robust NMF programs related to C5. The robust NMF program derived from the 3' workflow contained several non-coding RNA genes yet shared similar genes related to extracellular matrix organization, such as *TNC* and *P3H2* (Supplementary Fig. 8b). These results again highlighted the variations in transcriptomic coverage as the major difference between the Flex and the 3' assay inherited by the snPATHO-seq workflow. Nonetheless, all snRNA-seq workflows appeared to detect similar transcriptomic signatures, highlighting that the snPATHO-seq workflow can produce data with comparable transcriptomic features as conventional 10x 3' and Flex assays.

## The snPATHO-seq workflow offers more robust gene expression detection than the scFFPE workflow

In addition to the snPATHO-seq workflow, several other FFPE single-cell/nucleus RNA sequencing methods emerged during the preparation of this manuscript: snFFPE-seq[30], scFFPE[13], and snRandom-seq[31]. The snFFPE-seq study optimized procedures related to FFPE sample processing, including deparaffinization, tissue rehydration, and decross-linking, aiming to maximize the efficiency of RNA profiling using conventional single-cell gene expression readout assays such as SMART-Seq V2[32], SCRB-Seq[33] and 10x 3' assays on FFPE samples. However, the data generated using FFPE samples had about 2–2.7x lower RNA complexity than those generated using matching frozen tissue samples. This is likely because these gene expression readout assays rely on polyA-based capture and are less effective for FFPE samples with fragmented RNAs. As the authors highlighted, other gene expression readout assays using random primers and polyadenylation of small RNA fragments, such as Smart-seq-total[34], might circumvent this drawback[30].

This idea was reflected in the more recent snRandom-seq study, where the authors employed customized chemistry to amplify RNA fragments in FFPE samples using random primers[31]. The amplified cDNAs were then polyadenylated and captured using a microfluidic device for gene expression profiling[31]. The authors compared the performance of the snRandom-seq to snFFPE-seq and snPATHO-seq in terms of UMIs and genes detected per nucleus and demonstrated the superior performance of the snRandom-seq method[31]. As mentioned above, the 10x Flex chemistry only offers partial transcriptomic coverage, which applies to the snPATHO-seq workflow. Therefore, the differences in transcriptomic coverage could, to a certain extent, explain the differences between the snRandom-seq and the snPATHO-seq. In addition, these comparisons of common quality control metrics were made using datasets generated from different samples. A future study conducting a more direct comparison between the snRandom-seq and the snPATHO-seq using the same tissue samples can no doubt provide more insight into their variation in performance. Importantly, in a more recent study by the same research team, the authors automated the snRandom-seq workflow by integrating single-nucleus isolation and droplet barcoding platforms, significantly enhancing its throughput[35]. In a direct comparison with the manual snRandom-seq workflow, the automated workflow demonstrated a high degree of concordance with the manual workflow, highlighting the robustness of the automated approach[35]. This advancement marked a substantial improvement in both the throughput and standardization of snRandom-seq. Moreover, the commercialization of snRandom-seq by M20 Genomics increased its accessibility, making it more user-friendly and adaptable for broader adoption[36]. While further testing by external users will provide deeper insights into its performance, the snRandom-seq method shows great promise for advancing our understanding of FFPE single-nucleus research.

A fourth single-cell FFPE workflow, scFFPE, was released by 10x Genomics after the development of the snPATHO-seq workflow[13]. The scFFPE workflow shares many similarities with the snPATHO-seq workflow. Both workflows begin with tissue dewaxing and rehydration, followed by collagenase-based tissue dissociation and gene expression detected using the 10x Flex assay. However, instead of continuing with cell membrane lysis and nuclei extraction, as shown in the snPATHO-seq workflow, the scFFPE workflow used the dissociated cells directly for gene expression profiling[14]. Therefore, according to the developer, the scFFPE workflow was considered a single-cell RNA sequencing workflow instead of a single-nucleus RNA sequencing workflow[13], which constitutes the major difference between the snPATHO-seq and the scFFPE workflows. We favor nuclei isolation rather than cell isolation because nuclei can be more stably extracted from samples that are hard to dissociate into whole cells[37–39]. As highlighted in previous literature, snRNA-seq can often provide a more comprehensive cell type representation than scRNA-seq in certain types of tissues such as the brain, liver, and kidney[37–39].

To compare the performance of the scFFPE and snPATHO-seq workflows, we generated scFFPE data using the same breast cancer tissue samples (i.e., 4066, 4399, and 4411) mentioned above. The scFFPE workflow yielded incompletely dissociated tissue samples with mixtures of cells, nuclei, and tissue debris (Supplementary Fig. 3b). While we continued gene expression analysis using the prepared samples, the scFFPE workflow produced data with fewer UMIs and genes detected per cell and fewer cells detected in each sample than the snPATHO-seq method (Fig. 4a–c). It is

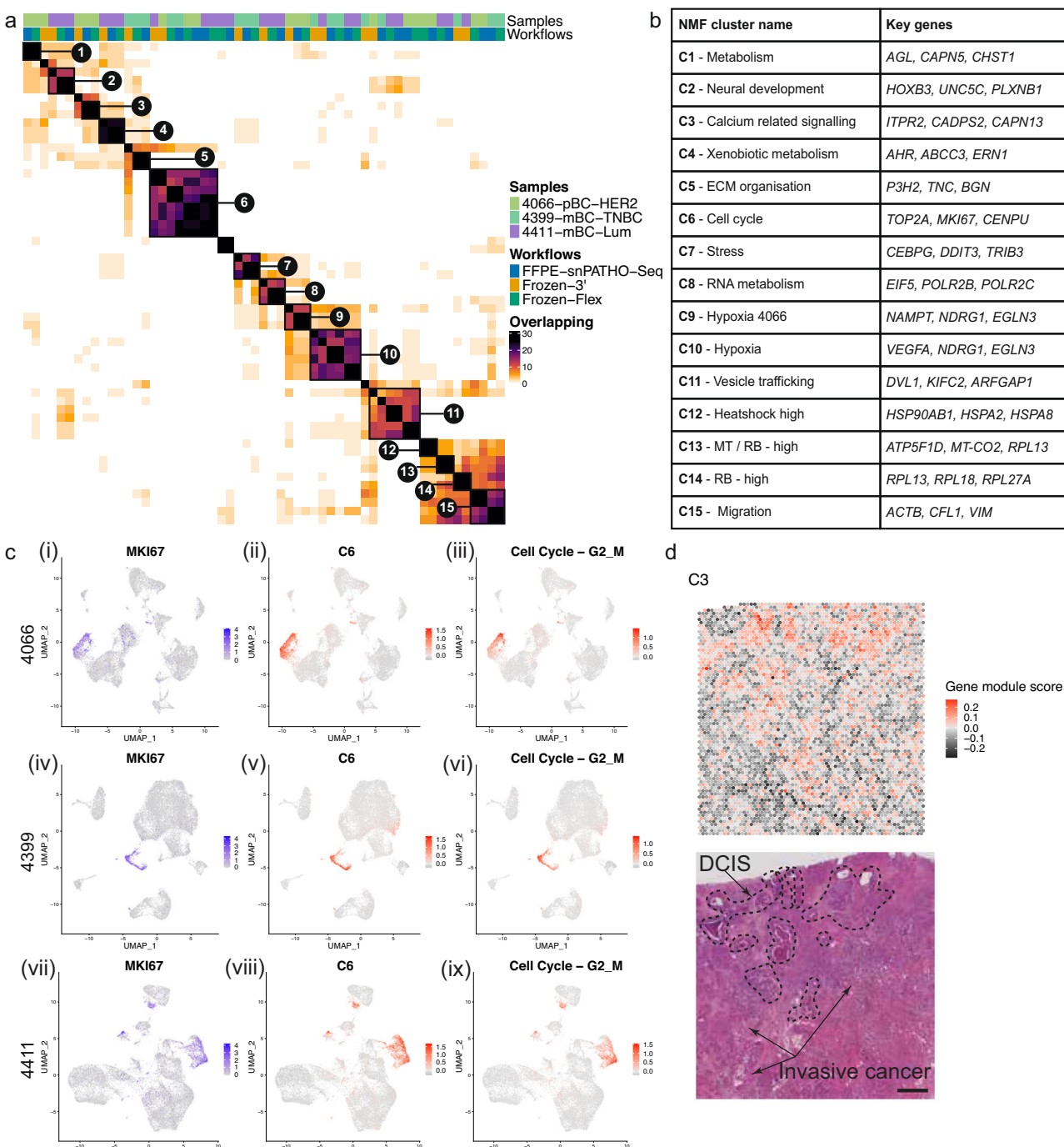

**Fig. 3 | snPATHO-seq detected comparable transcriptomic signatures from FFPE samples as the conventional Flex chemistry using matching snap-frozen samples. a** Heatmap showing similarities between robust NMF programs derived from snPATHO-seq, 10x 3', and 10x Flex data generated using 4066, 4399, and 4411. Robust NMF programs with similar gene compositions were clustered and highlighted in brackets. **b** Table of robust NMF program clusters annotated based on the shared gene compositions. **c** UMAP embedding of Seurat CCA integrated data from patients 4066 (i–iii), 4399 (iv–vi), and 4411 (vii–ix) overlaid with the expression of *MKI67* (i, iv, vii), the module scores of robust NMF program cluster C6 (ii, v, viii) and the module scores of a published cell cycle gene program[23] (iii, vi, ix). **d** Spatial enrichment pattern of the module score of robust NMF program cluster C3 in the Visium data from patient 4066 and the H&E image of this sample. Scale bar = 1 mm.

worth noting that a discrepancy in performance between the snPATHO-seq and the scFFPE workflow was also observed in a separate experiment conducted by 10x Genomics. In the colon Crohn's disease sample (Colon_1328A), while the scFFPE workflow was able to detect some macrophage and smooth muscle cells, it failed to capture the rest of the cell populations, including fibroblasts, endothelial cells, T cells, and B cells (Fig. 4d–f). Considering the overall similarities between the snPATHO-seq and the scFFPE workflows, the differences in performance likely stem from

the differences in single-cell/nucleus suspension preparation. In a recent technical update from 10x Genomics, the developers clarified the nature of scFFPE workflow in generating mostly nuclei instead of cells after tissue dissociation[14]. However, our evaluation highlighted the presence of both cells and nuclei in the dissociated tissue following scFFPE protocol. Therefore, the collagenase-based tissue dissociation alone was insufficient to prepare pure cells or nuclei suspensions from FFPE samples, at least in the breast cancer samples tested. A further nuclei extraction procedure is

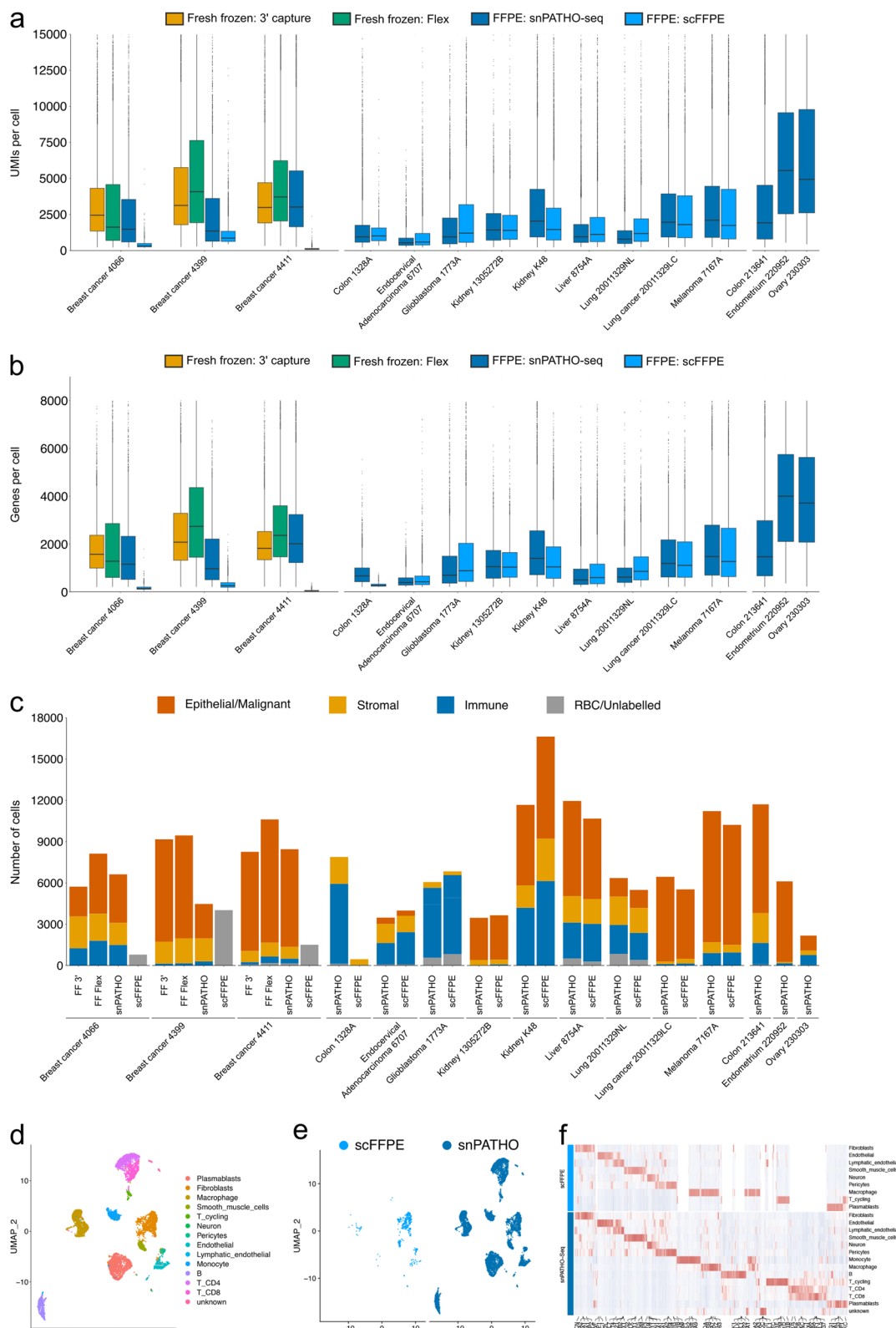

**Fig. 4 | Comparison of the snPATHO-seq workflow to scFFPE workflow.**
**a, b** Boxplots of the number of UMIs (**a**) and genes (**b**) detected per nuclei using different workflows. The boxes show the median and interquartile range of the UMIs (**a**) and Genes (**b**). Outliers were shown as dots. **c** Barplots of the number of nuclei/cells detected in each dataset colored by major cell lineages. **d** UMAP embedding of Seurat CCA integrated snPATHO-seq and scFFPE data from sample Colon_1328A colored by cell type annotations. **e** UMAP embedding split by processing methods. **f** Heatmap of top differentially expressed genes detected between cell types by snPATHO-seq and scFFPE workflows. The top 200 significantly differentially expressed genes identified in each cell population (if available) were selected by fold change and used for plotting. A gene was considered significantly differentially expressed if the BH-adjusted $P$ value was lower than 0.05. $N = 1$ sample per protocol.

necessary to standardize the dissociated samples into more uniformed single-nucleus suspensions as a more suitable input for the 10x Flex chemistry. In addition, as the nuclei extraction reagent can dissolve the cytoplasmic membrane, it likely functions as a clearing reagent, removing some tissue debris. The presence of tissue debris can lead to bias in cell counting and loading for droplet-based scRNA-seq analysis, which likely accounted for the failure of the gene expression data in the breast cancer samples tested[40].

Nonetheless, in many other FFPE tissue samples tested by 10x Genomics and the Fred Hutchinson Innovation Laboratory, the scFFPE could generate data with good concordance with the snPATHO-seq workflow (Fig. 4a–c, Supplementary Fig. 9). This highlighted that the scFFPE protocol was indeed effective against certain FFPE samples. Upon further investigation, we noticed that the breast cancer tissue samples tested were around 5–8 years old and slightly older than the other samples (0–5 years old) (Supplementary Data 1). It is known that the quality of RNA diminishes after storage of the FFPE blocks[41]. However, if and to what extent the age of the block affects the success of single-cell gene expression analysis remains to be further tested.

In this study, we developed a robust snRNA-seq workflow, snPATHO-seq, tailored to gene expression profiling of FFPE tissue samples. We benchmarked snPATHO-seq against 10x 3', Flex, and scFFPE workflows, demonstrating its robustness across a wide range of healthy and diseased tissue samples. Leveraging a multi-modal sampling strategy, snPATHO-seq can be applied to the same samples and integrated with the 10x FFPE Visium workflow, offering a more comprehensive transcriptomic characterization of FFPE samples. Notably, snPATHO-seq utilizes off-the-shelf reagents, resulting in low adaptation costs. We anticipate that the snPATHO-seq workflow will promote single-cell studies, especially in areas with limited sample availability. For instance, breast cancer metastases can happen years after the treatment of the primary disease and are less frequently biopsied. Incorporating the snPATHO-seq workflow into such projects could expand the size of the clinical cohort, thereby increasing the statistical power of these studies. However, transcriptomic coverage is a limitation of the snPATHO-seq workflow and is challenging to overcome. To this end, novel sample preservation methods such as FixNCut[42], which enables 10x 3' gene expression characterization, can be considered to improve the preservation of fresh tissue samples for multi-purposed genomic characterization. We believe that combining these various genomic technologies, each with unique advantages, can alleviate the constraints of sample availability in biological research. This integration can strengthen the connection between clinical and laboratory-based research, thereby enhancing the translational significance of these studies.

## Methods
### Patient material, ethics, and consent for publication
All ethical regulations relevant to human research participants were followed. For breast cancer samples, primary breast cancer sample 4066 was collected with written informed consent under the SVH 17/173 protocol with approval from St Vincent's Hospital Ethics Committee. Metastatic breast cancer samples 4399 and 4411 were collected as autopsy samples under protocol X19-0496. Consent included the use of all de-identified patient data for publication. Collected tumor samples were macroscopically dissected. For FFPE tissue preservation, samples were fixed in 10% neutral buffered formalin for 24 h at room temperature before being processed for paraffin embedding. For snap-frozen sample preservation, samples were diced into small chunks before snap-freezing in liquid nitrogen and stored at −80 °C.

For samples processed in the Cancer Research Centre of Lyon, the collection was provided by the 3D Onco platform (S. Ballesta) and Infirmerie Protestante de Lyon and was consented under the following ethical protocols (endometrium: 2022-05, colon: 2021-33, ovary: I-3422-01_MTA_IP_CLB) approved by the ethical review board of Centre Léon Bérard. The study was compliant with GDPR requirements and the CNIL (French National Commission for Computing and Liberties) law.

The other samples were obtained from commercial vendors, as documented (Supplementary Data 1).

### Single-cell RNA profiling for frozen PBMC samples
Two frozen PBMC aliquots from the same healthy donor were thawed and divided equally for processing using the standard 10X 3' and Flex assays, following the respective recommended protocols. For the 3' assay, the Chromium NextGEM Single-cell 3' Reagent kit (v3.1, 10x Genomics) was utilized as per the Chromium Single Cell 3' Reagent Kits User Guide (v3.1 Chemistry) (CG000204 - Rev D), targeting approximately 5000 cells. Meanwhile, for the Flex assay, PBMC cells underwent a brief fixation as directed by the Fixation of Cells & Nuclei for Chromium Fixed RNA Profiling guide (CG000478 - Rev A, 10x Genomics). Subsequently, the cells were profiled using the Chromium Fixed RNA Kit, Human Transcriptome (1000474, 10x Genomics), targeting a similar cell count as the 3' assay.

### Single-nucleus RNA profiling for snap-frozen tissue samples
For snap-frozen samples, nuclei were isolated as previously described[43] with modifications. Frozen samples were first thawed and finely minced. The fragments were subsequently homogenized using 1x Nuclei Ez lysis buffer (NUC-101, Sigma-Aldrich) enriched with 1 U/μL RNAse inhibitor (Ribo-Lock RNAse Inhibitor, EO0382, Thermo Fisher Scientific) and chilled for 5 min on ice. The separated nuclei were filtered through a 70 μm mesh (pluriStrainer 70 μm, 43-50070-51, pluriSelect) and rinsed twice with 1x PBS augmented with 1% BSA (MACS® BSA Stock Solution, 130-091-376, Miltenyi) and once with a 0.5x PBS + 0.02% BSA mix. They were subsequently resuspended in the 0.5x PBS + 0.02% BSA solution and re-filtered using a 40 μm mesh (pluriStrainer Mini 40 μm, 43-10040-50, pluriSelect). The final nucleus count was determined using the LUNA-FX7 cell counter (AO/PI viability kit, F23011, Logos).

For the 3' experiments, gene expression libraries were constructed using the Chromium NextGEM Single-cell 3' Reagent kit (v3.1, 10x Genomics), adhering to the Chromium Single Cell 3' Reagent Kits User Guide (v3.1 Chemistry) (CG000204 - Rev D), with a target of 10,000 nuclei per reaction.

For the Flex experiments, nuclei were first fixed as per the Fixation of Cells & Nuclei for Chromium Fixed RNA Profiling guide (CG000478 - Rev A, 10x Genomics). Gene expression libraries were then constructed using the Chromium Fixed RNA Kit, Human Transcriptome (1000474, 10x Genomics), based on the Chromium Fixed RNA Profiling user guide (CG000477 - RevB), targeting 10,000 nuclei for each reaction.

### Single-nucleus RNA profiling for FFPE tissue samples (snPATHO-seq)
Two tissue sections, each between 25 and 30 μm thick, were first washed thrice with 1 mL of Xylene for 10 min to remove paraffin. Subsequently, they were rehydrated via a sequence of 1 mL ethanol baths, each lasting 1 min: two rounds in 100% ethanol, then in 70%, 50%, and finally 30% ethanol. Specifically for breast tissues, the initial xylene wash was conducted at 55 °C. The sections were then briefly rinsed with 1 mL of RPMI1640 (Gibco).

Tissue disruption began physically using a pestle in 100 μL of a digestion mix composed of 1 mg/mL Liberase TM (5401119001, Roche), 1 mg/mL Collagenase D (11088858001, Roche), and 1 U/μL of RNAse inhibitor (RiboLock RNAse Inhibitor, EO0382, Thermo Fisher Scientific) in RPMI1640. This mix was then filled up to a 1 mL volume and subjected to digestion at 37 °C for 45–60 min.

For nuclear extraction, the pre-digested tissue was treated with 1x Nuclei Ez lysis buffer (NUC-101, Sigma-Aldrich) that included 2% BSA. Disintegration further proceeded via pipetting using a P1000 pipette. The separated nuclei were filtered through a 70 μm mesh, double-rinsed with 1x PBS supplemented with 1% BSA, and once with a 0.5x PBS + 0.02% BSA blend. They were then resuspended in this blend and re-filtered via a 40 μm mesh (pluriStrainer Mini 40 μm, 43-10040-50, pluriSelect). The concluding nuclear count was ascertained using the LUNA-FX7 cell counter (AO/PI viability kit, F23011, Logos).

Gene expression libraries were prepared using the Chromium Fixed RNA Kit, Human Transcriptome (1000474 or 1000475, 10x Genomics), in accordance with the user guide (Chromium Fixed RNA Profiling, CG000477 - RevB). We implemented additional optimizations in the pre-amplification and Indexing cycles tailored for nuclei derived from FFPE samples. Over 200,000 nuclei underwent a 20-h hybridization with the BC01 probe set. This was followed by three washes using the Post-Hyb Buffer as recommended in the guide, with an additional wash step. Post-hybridization, nuclei were resuspended in the Post-Hyb Resuspension Buffer, counted, and loaded onto Chip Q/Chromium X for capture, adhering to the guide's procedures. A 9-cycle pre-amplification was conducted. Indexing PCR cycles followed the guide's recommendations for PBMC and nuclei but with two added cycles. For sample 4399, the capture targeted 5000 nuclei, while for other samples, the target was set at 10,000 nuclei.

### Single-cell RNA profiling for FFPE tissue samples (scFFPE)
scFFPE workflow was conducted on the FFPE samples by the 10x Genomics and Fred Hutch Innovation Lab as outlined in Supplementary Data 1 following the demonstrated protocol (CG000632, RevB).

### Spatial transcriptomics on breast cancer samples
A 5-μm-thick section was prepared from the FFPE blocks and processed using the Visium Spatial Gene Expression for FFPE Kit v2 (10x Genomics) according to the manufacturer's instructions. Briefly, sections were H&E stained and imaged, followed by probe hybridization and ligation.

### Illumina sequencing
The indexed libraries were sequenced using the Illumina NextSeq550 or NovaSeq 6000 systems. For sc-/sn-RNAseq libraries, the read format was configured as 28, 10, 10, and 90 for Read 1, i7, i5, and Read 2 sequences, respectively. For Visium spatial transcriptomics libraries, the read format was configured as 28, 10, 10, and 50 for Read 1, i7, i5, and Read 2 sequences, respectively. Libraries were sequenced to a depth of more than 10,000 read pairs per cell/nuclei for sc-/sn-RNAseq experiments or more than 25,000 read pairs per spot for Visium experiments.

### FASTQ files processing
For data generated by the 10X Genomics, raw reads were de-multiplexed and aligned using Cellranger (v2023.0415.0). GRCh38 (build 2020-A, 10X Genomics) and Chromium Human Transcriptome Probe Set v1.0.1 were used as references.

For other single-cell/nucleus data, sequencing reads were de-multiplexed and aligned using Cellranger v7.0.1. GRCh38 (build 2020-A, 10X Genomics) was used as the reference for read mapping. For Flex-related workflows, Chromium Human Transcriptome Probe Set v1.0 reference was used for probe read mapping.

For Visium data, reads were processed using Spaceranger v2.0.0 with GRCh38 (build 2020-A, 10X Genomics) and Visium Human Transcriptome Probe Set v2.0 as references.

For direct comparison of different single-cell/nucleus workflows, the PBMC data was downsampled to ~30,000 reads per cell, and the breast cancer snRNA-seq datasets were downsampled to ~25,000 reads per nucleus. For 3' data, FASTQ files were downsampled using Seqtk (v1.3)[44]. For Flex assay data, FASTQ files were downsampled during Cellranger analysis by specifying the targeted numbers of reads per cell/nuclei in the configuration file supplied to the "cellranger multi" function.

### Ambient background filtering for snRNA-seq datasets
Additional ambient read filtering was conducted using Cellbender v0.2.0[45]. In total, 40,000 droplets were used to estimate the levels of background. The cell numbers estimated using the EmptyDrops method[46] implemented in the Cellranger were supplied as the expected number of cells for Cellbender processing.

### Low-quality data filtering for single-cell/nucleus RNA sequencing datasets
The count matrices from snRNA-seq datasets were then processed using Seurat (v4.3.0.9002)[16] in R (v4.1.1) unless otherwise specified. Low-quality cells/nuclei were defined as cells/nuclei with less than 200 UMIs, over 8000 UMIs, or over 10% mitochondrial gene products. Gene expression data was normalized using the "NormalizeData" function in the Seurat package. Doublets were identified using the DoubletFinder package (v2.0.3)[47] and excluded from the downstream analysis.

### Dimensionality reduction, clustering, and cell type annotation of sc/snRNA-seq data
For samples processed using more than one sc/sn-RNA workflow, the filtered counts were integrated at a per-sample level using the canonical correlation analysis (CCA) method as implemented in the Seurat package. The top 2000 highly variable genes in each dataset were used for the principal component analysis (PCA) and the top 2000 highly variable genes across datasets to be integrated were used to identify integration anchors.

For other datasets, PCA analysis was conducted using the top 2000 highly variable features.

For all datasets, Uniform Manifold Approximation and Projection (UMAP) processing was conducted using the top 30 principal components from the CCA or PCA analysis. Gene expression data was clustered using the top 30 principal components with the Louvain method as implemented in the Seurat package. Clusters were manually annotated based on the expression of canonical cell type markers. Re-integration or sub-clustering was also performed to separate cell lineages unable to be resolved by initial clustering.

### Differential gene expression analysis
Differential gene expression analysis was conducted between the annotated cell types on a per-sample basis with the "FindAllMarkers" function in the Seurat package. The Student's t-test was used, and all other parameters were kept as default. All data generated using different snRNA-seq workflows from the same sample was merged for this analysis.

To select top cell type markers for visualization, only genes with positive fold changes and BH-adjusted P values smaller than 0.05 were considered. The genes were then ranked by fold change in each cell type to select the top 200 genes (if applicable). The expression of the top genes was then scaled within each dataset (per sample, per snRNA-seq workflow) and visualized using the ComplexHeatmap package (v2.10.0)[48]. For breast cancer data, selected canonical cell type markers were plotted on the heatmaps. For all other datasets, the top 5 cell type markers by fold change were annotated on the heatmap.

### Automatic cell type annotation using singleCellNet
The published PBMC 3' dataset from Hao et al., 2021 was annotated using singleCellNet (v0.1.0)[49] and PBMC cell type signatures from this study. A classifier was trained using the top 50 genes from each cell type for 3' or Flex PBMC data, respectively. All other parameters were kept the same. The cell type annotation results were then visualized using the ggplot2 package (v3.4.2) included in tidyverse (v2.0.0) in R.

### Copy number variation inference
The copy number alteration of cancer cells was inferred using inferCNV (v1.10.1)[50] on a per-sample basis. All stromal and immune cell types within each dataset were selected as the reference cell types. Red blood cells (RBCs) and clusters with mixed cell type signatures were excluded from the dataset. For breast cancer 4399 datasets, as liver resident cell types were mainly identified in the snPATHO-seq dataset, they were also excluded from the reference. This includes cholangiocytes, hepatocytes, and LSECs. The analysis was then conducted on all normal and cancer epithelial cells with Hidden Markov Model (HMM) prediction. Default parameters were kept except for the cut-off, which was set to 0.1 according to the authors' recommendation for 10x Chromium data.

For breast cancer datasets, the HMM results for each gene were summarized across all cancer nuclei. The results were rounded up to the nearest multiple of 0.5. Any gene either not detected in the dataset or showed no CNV change was assigned with a value of 1. The results were summarized per dataset and plotted together using the ggplot2 package (v3.4.2) included in tidyverse (v2.0.0) in R.

## Spatial transcriptomics data processing

For Visium spatial transcriptomics data, tissue morphology was annotated at per spot level using the Loupe browser (v6.2.0, 10X Genomics). Spots underneath regions affected by processing artefacts were annotated as "exclude". The count metrics were converted into STutility objects (v1.1.1)[51] for downstream processing. Low-quality spots annotated as "exclude" were removed from the datasets. The metrics were then normalized using the "NormalizeData" function from the Seurat package implemented by the STutility package.

## Inference of spatial transcriptomics data cellular composition using deconvolution

Deconvolution of spatial cellular composition was performed using the RCTD method implemented in the spacexr package (v2.2.0)[52]. Datasets generated using 3', Flex, and snPATHO-seq were used as single-cell type references, respectively, for the deconvolution of matching Visium data. Default parameters were used for the analysis, except that the minimal number of cells required for each cell type was reduced to 3 to accommodate relatively low cell number in the current study. Deconvolution results were then visualized as heatmaps using the ComplexHeatmap package or as spatial feature plots using the ggplot2 package implemented in tidyverse.

## Robust NMF program identification and downstream analysis

Robust NMF programs were derived using a published method[23] with the NMF package (v0.26) installed in R (v4.2.3). NMF analysis was first conducted on breast cancer snRNA-seq data on a per dataset (per sample, per snRNA-seq workflow) basis using ranks from 4 to 9 with 10 iterations at each rank. This resulted in a total of 39 NMF programs derived from each dataset. The NMF programs from all datasets were then filtered based on similarities in gene composition using the published thresholds to define robust NMF programs[23]. The robust NMF programs were then clustered based on similarities to derive the final robust NMF program clusters using the clustering method derived by the original authors[23].

To evaluate the enrichment patterns of the robust NMF program clusters in snRNA-seq and Visium data, Seurat gene module scores were calculated using the "AddModuleScore" function with genes shared between at least two robust NMF programs in each cluster[53].

## Reporting summary

Further information on research design is available in the Nature Portfolio Reporting Summary linked to this article.

## Data availability

Raw and processed snRNA-seq and Visium spatial transcriptomics data have been uploaded to Gene Expression Omnibus (GEO) under the accession codes GSE268426 and GSE268427, respectively. Published PBMC data was downloaded from the GEO through the accession code GSE164378. Processed snRNA-seq and Visium data objects can be accessed through the Dryad repository (https://doi.org/10.5061/dryad.7m0cfxq4s)[54].

## Code availability

Scripts used to analyze snRNA-seq and Visium spatial transcriptomics data can be accessed at https://github.com/TaopengWang/snPATHO-seq_public or Zenodo[55].

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

## Acknowledgements

This work was supported by the National Health and Medical Research Council (NHMRC) Ideas Grant (APP2004774), the Commonwealth Standard Grant Agreement (4-F26M8TZ), and the generosity of J. McMurtrie, AM and D. McMurtrie, and the Petre Foundation. We thank the Institut PLAsCAN and "La ligue contre le cancer – Comité de l'Ain" for samples processed in the Cancer Research Center of Lyon. A.S. is the recipient of an investigator grant from the NHMRC (APP1161216) and is a Breast Cancer Research Foundation (BCRF) investigator. T.W. was supported by an Australian Government Research Training Program Scholarship. The South Australian immunoGENomics Cancer Institute (SAiGENCI) received grant funding from the Australian Government. The funders had no role in study design, data collection, analysis, decision to publish, or preparation of the manuscript. We thank Jens Durruthy Durruthy, Daniel "Telstra" Dlugolensky, Andrew Kohlway from 10x Genomics, and the team at Millennium Sciences for the technical support and for providing access to the 10x Genomics Flex kit. We thank Naiara Bediaga and Monika Mohenska from the University of Adelaide for facilitating data organization. We thank Garvan Histology for fast turnover tissue processing.

## Author contributions

Conceptualization: Alexander Swarbrick (A.S.), Luciano G. Martelotto (L.G.M.); Formal analysis: Taopeng Wang (T.W.), Michael J. Roach (M.J.R.); Funding acquisition: L.G.M. and A.S.; Data curation, resources, investigation, methodology, software, validation: T.W., M.J.R., Kate Harvey, Anna Elz, Azi Hadadianpour, Jack Johanneson, Fiona Pakiam, Heeju Ryu, Evan W. Newell, Andrew Kohlway, Tingsheng Drennon, Jawad Abousoud, Ryan Stott, Paul Lund, Jens Durruthy, Andres F. Vallejo, Wenyan Li, Robert Salomon; Javier Escudero Morlanes, Alissa Greenwald, Yuling Ma, Yered H. Pita-Juarez, Cyril Degletagne, Nikolaos Kalavros, Felipe Segato Dezem, Kellie Wise, Laurie Tonon, Jasmine Plummer, Ioannis S. Vlachos, Joakim Lundeberg, T.W., Beata Kiedik, Ghamdan Al-Eryani, Dominik Kaczorowski, Joanna Warren, Lisa M. Butler, Sandra O'Toole, and L.G.M.; Project administration: T.W., L.G.M., and A.S.; Supervision: A.S. and L.G.M.; Visualization: T.W., M.J.R., K.W., and L.G.M.; Writing—original draft preparation: T.W., M.J.R., A.S., and L.G.M.; Writing—review & editing: all authors.

## Competing interests

The authors declare no competing interests.

## Additional information

[1]Cancer Ecosystems Program, Garvan Institute of Medical Research, Darlinghurst, NSW, Australia. [2]School of Clinical Medicine, Faculty of Medicine and Health, UNSW Sydney, Sydney, NSW, Australia. [3]Adelaide Centre for Epigenetics, University of Adelaide, Adelaide, SA, Australia. [4]South Australian Immunogenomics Cancer Institute, University of Adelaide, Adelaide, SA, Australia. [5]College of Science and Engineering, Flinders University, Adelaide, SA, Australia. [6]KTH Royal Institute of Technology, Science for Life Laboratory, Solna, Sweden. [7]Department of Molecular Cell Biology, Weizmann Institute of Science, Rehovot, Israel. [8]Department of Pathology, Beth Israel Deaconess Medical Center, Boston, MA, USA. [9]Harvard Medical School, Boston, MA, USA. [10]Broad Institute of MIT and Harvard, Cambridge, MA, USA. [11]Spatial Technologies Unit, Harvard Medical School Initiative for RNA Medicine, Beth Israel Deaconess Medical Center, Boston, MA, USA. [12]Center for Spatial Omics, St. Jude Children's Research Hospital, Memphis, TN, USA. [13]CRCL Core facilities, Centre de Recherche en Cancérologie de Lyon (CRCL) INSERM U1052-CNRS UMR5286, Université de Lyon, Université Claude Bernard Lyon, Centre Léon Bérard, Lyon, France. [14]Fred Hutch Innovation Lab, Fred Hutchinson Cancer Center, Seattle, WA, USA. [15]Vaccine and Infectious Disease Division, Immunotherapy Integrated Research Center, Fred Hutchinson Cancer Center, Seattle, WA, USA. [16]Fondation Synergie Lyon Cancer, Plateforme de Bioinformatique Gilles Thomas, Centre Léon Bérard, Lyon, France. [17]10X Genomics, Pleasanton, CA, USA. [18]Clinical and Experimental Sciences, Sir Henry Wellcome Laboratories, Faculty of Medicine, University of Southampton, Southampton, UK. [19]Children's Cancer Institute, UNSW Lowy Cancer Research Centre, Kensington, NSW, Australia. [20]School of Clinical Medicine, UNSW Medicine & Health, UNSW Sydney, Kensington, NSW, Australia. [21]Cellular Genomics Platform, Garvan Institute of Medical Research, Sydney, NSW, Australia. [22]Solid Tumour Program, Precision Cancer Medicine Theme, South Australian Health and Medical Research Institute, Adelaide, SA, Australia. [23]Department of Tissue Pathology and Diagnostic Oncology, NSW Health Pathology, Royal Prince Alfred Hospital, Sydney, NSW, Australia. [24]Sydney Medical School, University of Sydney, Sydney, NSW, Australia. [25]School of Medicine, University of Western Sydney, Sydney, NSW, Australia. [26]These authors contributed equally: Taopeng Wang, Michael J. Roach. [27]These authors jointly supervised this work: Alexander Swarbrick, Luciano G. Martelotto. ✉e-mail: a.swarbrick@garvan.org.au; luciano.martelotto@adelaide.edu.au

