## [Transparent Peer Review file · Communications Biology]

snPATHO-seq, a versatile FFPE single-nucleus RNA sequencing method to unlock pathology archives

Corresponding Author: Dr Luciano Martelotto

Version 0:

Reviewer comments:

Reviewer #1

(Remarks to the Author)

Thank you for inviting me to evaluate the article titled "snPATHO-seq: unlocking the pathology archives". This paper proposes a new method, snPATHO-seq, to derive high-quality single-nucleus transcriptomic data from FFPE samples. Although this work is of interest in the field, there are certain issues that should be addressed in the next version. The manuscript is not well arranged and the logic is not clear. In the abstract section, the author needs to outline what is the benefit of this method and the findings from using snPATHO-seq on diverse clinical specimens.

The introduction could be expanded to include pertinent scientific history on single-cell/nucleus RNA sequencing (scRNA/snRNA-seq) techniques.

The chemistry of this technique, which used the commercial Flex from 10x Genomics, is not improved by snPATHO-seq. Thus, the only possible addition could be a novel approach to the production of FFPE samples; however, this point is not elaborated properly.

There should be more in the discussion area. This new method has to be compared with other methods in the discussion section even though the authors evaluated the performance of snPATHO-seq on FFPE against other snRNA-seq procedures in the result description. It is recommended that the authors emphasize the advantages of snPATHO-seq and include afterthought of this work in the discussion section.

Given that the nucleus has a diameter of around 20 μm , what proportion of fragmented nuclei are there in the isolated samples when tissue sections with a thickness of 25–30 μm are used?

Reviewer #2

(Remarks to the Author)

Wang and colleagues describe a new approach, snPATHO-seq, which aims to perform single-cell sequencing with formalin-fixed paraffin-embedded tissue (FFPE). Hospitals routinely use formalin fixing and paraffin embedding for their clinical samples, such as tumors and other histopathological assays. Therefore, there is a wealth of information stored in the biosample banks. Profiling FFPE with bulk or, more recently, single-cell transcriptomics is challenging because formalin is a fixative that severely affects RNA integrity. There is a clear need for approaches that offer a general solution for single-cell profiling of FFPE tissue.

The authors address this by developing snPATHO-seq, an approach that uses the 10X Genomics Flex system. This system uses RNA probes and is therefore more robust to low RNA integrity. The authors first apply their approach to gold standard PBMCs and then to a wide range of other clinical cancer and normal tissues. They demonstrate that snPATHO-seq works in various scenarios and is comparable to other 10X Genomics technologies. Therefore, snPATHO-seq is a robust technique, and its publication would allow other researchers to harness the approach.

However, there are several shortcomings that the authors could address prior to publication. Furthermore, the paper is very short, and at times, it would benefit from additional explanations.

Concerning Flex vs. snPATHO-seq, the authors use Flex probes in their snPATHO-seq approach, but it is unclear from the text what Flex precisely is, what snPATHO-seq is, and what the differences are between the two. The authors might want to explain this in better detail and perhaps provide a cartoon. Figure 1a ("snPATHO-Seq workflow") is quite shallow and devoid of details. Here, the paper would improve if a more technical workflow cartoon, briefly enumerating the steps and perhaps highlighting the steps that differ between their approach and general Flex, were included. This could also be further discussed in the text.

Later in the text, the authors acknowledge that 10X Genomics has introduced an scFFPE protocol, which I assume is largely equivalent to snPATHO-seq. I agree with the authors in pursuing the publication of snPATHO-seq as an open protocol, as the version by 10X Genomics will likely be under IP protection and it is worth having this in the public domain. But this makes the point described above even more crucial: the authors should clarify what the differences are between snPATHO-seq and the original Flex approach and how similar the new approach introduced by 10X Genomics is to their approach.

The authors later comment that the 10X approach was ineffective in some samples. While the authors say that the reasons why this happened are unclear, they might want to offer more information or comment on it in greater length to help the reader decide whether this is just a quirk or a systematic flaw.

The major aim of the authors is to benchmark snPATHO-seq and show that it provides quality and stats similar to other methods and that the data integrates well. However, integration methods such as CCA are specifically designed to soften differences, assuming that they are batch effects. In essence, there is a conceptual loophole here: the conclusion is that the data looks similar after applying a method that is designed to make the data similar. Obviously, the authors would not be able to integrate the data if there were major differences, and I believe that the data indeed looks comparable and can be integrated. But the paper and the comparison would improve if the authors explored a little bit what happens in different integration conditions and/or methods. This could help find the differences, instead of the commonalities, between the different methods. Such an exercise would improve the paper and provide more fine-grain information. For instance, the authors use 2000 PCs for their analyses and the CCA anchor identification, and that might well be enough for these individual datasets. However, it is a possibility that on the top 2000 PCs, similarities dominate, but that differences between methods start to be seen in subsequent PCs. Of course, this would indicate that these are minor differences, but still worth reporting. As opposed to this notion, it is possible that the authors are able to integrate the datasets using more anchors as well and cannot identify differences. Either way, the results of such a parameter exploration would improve the paper.

I am, of course, not demanding that the authors explore all parameters in all of their analyses, as this would prove to be an incredible amount of work. However, the authors could select a few parameters to examine and test them in a few examples.

Minor:

The authors might want to provide rasterized figures of their UMAPs in the future. That would avoid long PDF loading and printing time. Of course, upon acceptance, they might want to provide full-resolution images, but a rasterized UMAP image embedded in the PDF would make things lighter and easier at this stage. Both Seurat and Scanpy can generate PDF plots with rasterized dots.

Reviewer #3

(Remarks to the Author)

This manuscript addresses an important application of scRNA-seq technologies to the clinical setting. The authors evaluate a novel strategy of single-cell gene quantification by RNA-targeting probes (Flex by 10X Genomics). This approach is highly applicable to clinically widely-accessible FFPE samples, which commonly show high levels of RNA fragmentation. The authors developed snPATHO-seq, an snRNA-seq workflow tailored for FFPE tissue samples and state that it has comparable performance to other snRNA-seq workflows, including scFFPE by 10X Genomics.

The study is experimentally and computationally well-performed, and its conclusions would be highly valuable for the scientists in the field and to a wider audience.

Here are my concerns and suggestions.

Major comments:

The authors report an unclear discrepancy when comparing snPATHO-seq to scFFPE, and show a much lower performance of scFFPE (Ext Data Fig. 2). This result was only obtained for breast cancer tissue, while the comparison across multiple tissues did not show any dramatic differences. According to Supplemental Table S1, the breast cancer tissue experiment was only performed once on 3 patients at a single time and the samples were collected at a specific site. Therefore, there might be several covariates influencing the result, which makes the conclusion made by the authors uncertain. I suggest that this experiment should be repeated several times if the authors wish to make the claim that snPATHO-seq is superior to scFFPE for breast cancer samples.

Similarly, the rationale behind favoring nuclear rather than whole-cell preparations for snPATHO-seq is not well supported. Ext Data Fig. 4 does not quantitatively assess the whole-cell to nucleus ratio. I suggest that the authors show some additional results, such as quantification of nuclear and cytosolic protein markers by Western analysis of cellular/nuclear extracts.

The authors should list the main benefits for using snPATHO-seq over other more readily available protocols. It appears that the procedure is not easier, rather more complex than scFFPE, since there is an additional nuclear extraction step. Also, please include the composition of the nuclear extraction buffer and other details of the experimental protocol, since this is extremely beneficial for experimentalists looking to use snPATHO-seq in their work.

There were at least two more highly similar methods published recently, which should be compared to snPATHO-seq; snRandom-seq (<https://www.nature.com/articles/s41467-023-38409-5#ref-CR15>) and snFFPE-seq

(<https://www.biorxiv.org/content/10.1101/2022.08.25.505257v1>). In light of this, the novelty of snPATHO-seq should be presented in the manuscript. To strengthen the conclusions of the current manuscript, kidney snPATHO-seq data could be directly compared to the snRandom-seq results.

How reproducible are cell type fractions within the same sample and between different workflows? This would be important to show in Fig.1a to show reproducibility between snPATHO-seq, Frozen-3' and Frozen-Flex. Apart from absolute numbers (presented in Extended Data Fig.2c), scaled cell fractions would allow easier comparisons between workflows.

Minor comments:

Fig.1c only shows the UMAP result from down-sampled data. Can the clustering results obtained with whole data be shown in the supplement?

Ext Data Fig.1f&g: Instead of pairwise overlaps, the confidence scores of the label transfer would be more informative to assess the cell type annotation quality.

Abstract could be longer and could emphasize the impact of the paper.

Paper titles could be more informative and search friendly.

Author Rebuttal letter:

Response to Reviewers

Overview

We thank the Reviewers for their thoughtful feedback and insightful comments in helping us significantly improve our manuscript. We first summarize the key aspects of our revision and then provide a point-by-point response to each Reviewer's comments.

Key highlights

1. Comparison of the snPATHO-seq workflow to 10x 3' and Flex methods: Following the Reviewers' suggestions, we have included additional descriptions on the snPATHO-seq workflow to clarify its difference from the conventional 10x Flex workflow (Introduction & Results and Discussion section 2). The data generated using each method was also compared without the use of data integration (i.e. PCA on unintegrated PBMC and breast cancer data & NMF analysis) (Results and Discussion section 1-3).

2. Comparison of the snPATHO-seq workflow to other FFPE single-cell/nucleus RNA sequencing methods: Following the Reviewers' suggestions, we have included a more extensive discussion and comparison between the snPATHO-seq workflow and other FFPE single-cell RNA sequencing methods (i.e. snFFPE-seq, snRandom-seq and scFFPE) on data quality and usability (Results and Discussion section 4).

3. Manuscript formatting: The manuscript has been expanded to provide more information on the history of single-cell RNA sequencing (Introduction), key features of the snPATHO-seq workflow (Results and Discussion section 2), and major differences between the snPATHO-seq and other FFPE snRNA-seq workflows (Results and Discussion section 4).

Response to Reviewer 1:

Thank you for inviting me to evaluate the article titled "snPATHO-seq: unlocking the pathology archives". This paper proposes a new method, snPATHO-seq, to derive high-quality single-nucleus transcriptomic data from FFPE samples. Although this work is of interest in the field, there are certain issues that should be addressed in the next version. The manuscript is not well arranged and the logic is not clear. In the abstract section, the author needs to outline what is the benefit of this method and the findings from using snPATHO-seq on diverse clinical specimens.

Thank you for your kind words and for recognizing the importance of our work. We have updated the abstract to summarise our findings and the significance of snPATHO-seq for analyzing clinical specimens.

The introduction could be expanded to include pertinent scientific history on single-

cell/nucleus RNA sequencing (scRNA/snRNA-seq) techniques.

The chemistry of this technique, which used the commercial Flex from 10x Genomics, is not improved by snPATHO-seq. Thus, the only possible addition could be a novel approach to the production of FFPE samples; however, this point is not elaborated properly.

We thank the reviewer for this excellent suggestion. The manuscript has now been expanded to highlight the differences between the snPATHO-seq and other FFPE single-cell technologies (Results and Discussions section 4). Specifically, the main difference between the snPATHO-seq and the scFFPE workflow is the additional nuclei isolation step employed in the snPATHO-seq method. Based on our evaluation (Results and Discussions section 4), enzyme-based tissue dissociation alone was insufficient in producing pure single-cell/nucleus suspension for downstream gene expression analysis. The nuclei isolation step serves as a standardization process to ensure the capture of uniformed nuclei instead of mixtures of nuclei and cells by the downstream droplet-based capturing method.

There should be more in the discussion area. This new method has to be compared with other methods in the discussion section even though the authors evaluated the performance of snPATHO-seq on FFPE against other snRNA-seq procedures in the result description. It is recommended that the authors emphasize the advantages of snPATHO-seq and include afterthought of this work in the discussion section.

We thank the reviewer for this excellent suggestion. The results and discussion section (Results and Discussions section 4) has been expanded to compare the snPATHO-seq method to the other existing methods, including snFFPE-seq, snRandom-seq, and scFFPE. Afterthought was included in the conclusion section.

Given that the nucleus has a diameter of around 20 μm , what proportion of fragmented nuclei are there in the isolated samples when tissue sections with a thickness of 25–30 μm are used?

We thank the reviewer for this question. For the breast cancer samples tested, quantification of the H&E image suggested that nuclei size ranges from 12-14 μm in diameter (example image below). Therefore, many nuclei are expected to be intact after sectioning. While some degrees of fragmentation is expected, the microscopic images of the samples after snPATHO-seq processing revealed a minimal amount of tissue debris. The fragmented nuclei were likely removed together with other tissue debris during washing.

Response to Reviewer 2:

Wang and colleagues describe a new approach, snPATHO-seq, which aims to perform single-cell sequencing with formalin-fixed paraffin-embedded tissue (FFPE). Hospitals routinely use formalin fixing and paraffin embedding for their clinical samples, such as tumors and other histopathological assays. Therefore, there is a wealth of information stored in the biosample banks. Profiling FFPE with bulk or, more recently, single-cell transcriptomics is challenging because formalin is a fixative that severely affects RNA integrity. There is a clear need for approaches that offer a general solution for single-cell profiling of FFPE tissue.

The authors address this by developing snPATHO-seq, an approach that uses the 10X Genomics Flex system. This system uses RNA probes and is therefore more robust to low RNA integrity. The authors first apply their approach to gold standard PBMCs and then to a wide range of other clinical cancer and normal tissues. They demonstrate that snPATHO-seq works in various scenarios and is comparable to other 10X Genomics technologies. Therefore, snPATHO-seq is a robust technique, and its publication would allow other researchers to harness the approach.

Thank you for your kind words and excellent summary of our work.

However, there are several shortcomings that the authors could address prior to publication. Furthermore, the paper is very short, and at times, it would benefit from additional explanations.

Concerning Flex vs. snPATHO-seq, the authors use Flex probes in their snPATHO-seq approach, but it is unclear from the text what Flex precisely is, what snPATHO-seq is, and what the differences are between the two. The authors might want to explain this in better detail and perhaps provide a cartoon. Figure 1a ("snPATHO-Seq workflow") is quite shallow and devoid of details. Here, the paper would improve if a more technical workflow cartoon, briefly enumerating the steps and perhaps highlighting the steps that differ between their approach and general Flex, were included. This could also be further discussed in the text.

Thank you for this excellent suggestion. We have expanded the manuscript to clarify the differences between Flex and snPATHO-seq workflows. Figure 2 now includes a more informative workflow illustration of the snPATHO-seq workflow. The more detailed snPATHO-seq protocol and a printable quick-reference workflow have been released online at protocols.io (<https://www.protocols.io/view/snpatho-seq-8epv5x58dg1b/v1> and https://figshare.com/articles/figure/Printable_workflow_protocol_for_snPATHO-seq_v1_/25210913?file=44520059).

Later in the text, the authors acknowledge that 10X Genomics has introduced an scFFPE protocol, which I assume is largely equivalent to snPATHO-seq. I agree with the authors in pursuing the publication of snPATHO-seq as an open protocol, as the version by 10X Genomics will likely be under IP protection and it is worth having this in the public domain. But this makes the point described above even more crucial: the authors should clarify what the differences are between snPATHO-seq and the original Flex approach and how similar the new approach introduced by 10X Genomics is to their approach.

We thank the reviewer for this comment. In the Results and Discussion section 4, we have now included more description of the differences between the snPATHO-seq and the scFFPE workflow.

The authors later comment that the 10X approach was ineffective in some samples. While the authors say that the reasons why this happened are unclear, they might want to offer more information or comment on it in greater length to help the reader decide whether this is just a quirk or a systematic flaw.

We thank the reviewer for this comment. The Results and Discussion section 4 has been updated to highlight the discrepancy in performance between the two workflows in the colon Crohn's disease dataset generated by the 10x Genomics. Considering that such discrepancy was observed in two datasets (breast & colon) generated by different teams, we reasoned that this is unlikely to be a random event but represents a more systematic issue of the scFFPE method. We also highlighted the recent update from the 10x Genomics, stating that the scFFPE produces nuclei instead of cells. This statement echoes our previous observation, suggesting that the scFFPE workflow is less stable than the snPATHO-seq and may struggle to generate high-quality data in certain samples. However, since the aim of this study is not to optimize the scFFPE method and we have limited sample and funding availability, further studies are needed to evaluate the performance of the scFFPE method more systematically.

The major aim of the authors is to benchmark snPATHO-seq and show that it provides quality and stats similar to other methods and that the data integrates well. However, integration methods such as CCA are specifically designed to soften differences, assuming that they are batch effects. In essence, there is a conceptual loophole here: the conclusion is that the data looks similar after applying a method that is designed to make the data similar. Obviously, the authors would not be able to integrate the data if there were major differences, and I believe that the data indeed looks comparable and can be integrated. But the paper and the comparison would improve if the authors explored a little bit what happens in different integration conditions and/or methods. This could help find the differences, instead of the commonalities, between the different methods. Such an exercise would improve the paper and provide more fine-grain information. For instance, the authors use 2000 PCs for their analyses and the CCA anchor identification, and that might well be enough for these individual datasets. However, it is a possibility that on the top 2000 PCs, similarities dominate, but that differences between methods start to be seen in subsequent PCs. Of course, this would indicate that these are minor differences, but still worth reporting. As opposed to this notion, it is possible that the authors are able to integrate the datasets using more anchors as well and cannot identify differences. Either way, the results of such a parameter exploration would improve the paper.

I am, of course, not demanding that the authors explore all parameters in all of their analyses, as this would prove to be an incredible amount of work. However, the authors could select a few parameters to examine and test them in a few examples. We thank the reviewer for this insightful comment and suggestions. We have now included Figure S1 and Figure 2d,e to highlight the variations between the 3' and Flex assays. Essentially, the Flex assay has reduced transcriptomic coverage compared to the 3' assay. This variation separates the 3' and Flex data in the lower dimension space. Since the snPATHO-seq workflow also uses the Flex assay for gene expression analysis, the snPATHO-seq data also showed substantial variation compared to the 3' data while displaying more similarities with the Flex assay data using matching frozen tissue samples. Therefore, an integration-based approach allowed us to identify similar cell types detected

by different snRNA-seq methods to compare the gene expression features between similar cell types.

Additionally, while our comparisons were based on an integration-driven approach, this approach did not seem to fully eliminate the variations between workflows. As highlighted in the 4399 dataset, the 3' and the Flex workflow failed to detect liver tissue-resident cell types using snap-frozen samples, while these cell populations were identified in the snPATHO-seq dataset using the FFPE sample. This highlighted that our approach could reveal variations between datasets, at least at the cell type level.

In addition, to more systematically and unbiasedly illustrate the variations between the workflows, we expanded the discussions around the NMF-based analysis results (Results and Discussion section 3). Despite utilizing some of the conclusions from integration (i.e., to identify cancer cells), this method operates on a per-dataset basis, allowing us to generate biologically meaningful NMF programs from each sample processed using each snRNA-seq method. We highlighted that the NMF analysis generally extracted similar transcriptomic signals from the Flex and snPATHO-seq datasets with high similarity in gene composition (Figure 3a). While many NMF programs extracted from the 3' data also showed high similarity to those extracted from the Flex and snPATHO-seq data using matching samples (Figure 3a), the variations in transcriptomic coverage contributed to variations between these results. This was demonstrated using the calcium-signaling-related and the ECM organisation related signatures, where robust NMF programs derived using the snPATHO-seq and Flex methods share more similarities with each other, while the robust NMF programs derived from the 3' workflow contain many non-coding RNAs and genes that were not probed for in the Flex assay. Our analysis again highlighted a major difference between the 3', Flex, and snPATHO-seq methods in transcriptomic coverage due to variations in assay design.

Minor:

The authors might want to provide rasterized figures of their UMAPs in the future. That would avoid long PDF loading and printing time. Of course, upon acceptance, they might want to provide full-resolution images, but a rasterized UMAP image embedded in the PDF would make things lighter and easier at this stage. Both Seurat and Scanpy can generate PDF plots with rasterized dots.

Thank you for this suggestion. We have rasterized some figures to improve loading times.

Response to Reviewer 3:

This manuscript addresses an important application of scRNA-seq technologies to the clinical setting. The authors evaluate a novel strategy of single-cell gene quantification by RNA-targeting probes (Flex by 10X Genomics). This approach is highly applicable to clinically widely-accessible FFPE samples, which commonly show high levels of RNA fragmentation. The authors developed snPATHO-seq, an snRNA-seq workflow tailored for FFPE tissue samples and state that it has comparable performance to other snRNA-seq workflows, including scFFPE by 10X Genomics.

The study is experimentally and computationally well-performed, and its conclusions would be highly valuable for the scientists in the field and to a wider audience.

Here are my concerns and suggestions.

Thank you for your kind words and excellent summary of our work.

Major comments:

The authors report an unclear discrepancy when comparing snPATHO-seq to scFFPE, and show a much lower performance of scFFPE (Ext Data Fig. 2). This result was only obtained for breast cancer tissue, while the comparison across multiple tissues did not show any dramatic differences. According to Supplemental Table S1, the breast cancer tissue experiment was only performed once on 3 patients at a single time and the samples were collected at a specific site. Therefore, there might be several covariates influencing the result, which makes the conclusion made by the authors uncertain. I suggest that this experiment should be repeated several times if the authors wish to make the claim that snPATHO-seq is superior to scFFPE for breast cancer samples.

We thank the reviewer for the insightful comments. We want to highlight the colon Crohn's disease dataset generated by the 10x Genomics, demonstrating the discordance between the snPATHO-seq and scFFPE workflows. In this dataset, the snPATHO-seq successfully captured a diverse range of cell types, while the scFFPE workflow predominantly detected smooth muscle cells and fibroblasts. We noticed that this point was

not well elaborated in the previous manuscript and have now improved the manuscript accordingly (Results and Discussion section 4). Nonetheless, we acknowledge that the scFFPE workflow performed well across various healthy and diseased human tissue samples tested by the 10x Genomics and the Fred Hutchinson Innovation laboratory. Based on our investigation using breast cancer tissue samples, the variation in performance is likely due to the variations in tissue dissociation between these workflows. While we were able to demonstrate the difference in the quality of the tissue dissociation between snPATHO-seq and scFFPE using breast cancer samples, we do not have images of the tissue dissociation products from other experiment sites (i.e., 10x Genomics and Fred Hutchinson Innovation laboratory). In addition, due to limitations in sample and funding availability, we could not repeat this experiment for comparison. However, despite these limitations, the fact that such discordance was observed twice from two different institutes suggested that our observation is unlikely to be a random event. In addition, 10x Genomics has recently amended the scFFPE protocol to highlight its nuclei-isolating nature. This raised some questions about the scFFPE workflow: what does the tissue dissociation product look like using the scFFPE workflow? Why was it named as a single-cell RNA sequencing workflow while the data generated were indeed nuclei? Could the generation of cells or nuclei be sample-specific? Therefore, the scFFPE workflow is still under active development, while the snPATHO-seq method is more mature. Since this manuscript aims to establish the snPATHO-seq as an FFPE snRNA-seq method, optimizing the scFFPE workflow is not a major focus of this study. Nonetheless, the snPATHO-seq workflow has been released online (<https://www.protocols.io/view/snpatho-seq-8epv5x58dg1b/v1> and https://figshare.com/articles/figure/Printable_workflow_protocol_for_snPATHO-seq_v1_/25210913?file=44520059) and we are keen for interested researchers to conduct a more systematic comparison to the scFFPE workflow to better understand the scFFPE method.

Similarly, the rationale behind favoring nuclear rather than whole-cell preparations for snPATHO-seq is not well supported. Ext Data Fig. 4 does not quantitatively assess the whole-cell to nucleus ratio. I suggest that the authors show some additional results, such as quantification of nuclear and cytosolic protein markers by Western analysis of cellular/nuclear extracts.

We thank the reviewer for this suggestion. We favored nuclei isolation since it is more stable than cell isolation for samples that are challenging to dissociate. This was mainly based on prior knowledge from fresh/frozen brain/kidney/liver tissue samples published in the literature and our experience working with FFPE samples. We have expanded the manuscript to incorporate these comments.

We attempted to provide a more quantitative assessment of the cells:nuclei ratio. However, as mentioned above, many flow cytometry markers underperform due to the formalin-induced crosslinking in FFPE samples. Reversing the crosslinking using heat and enzymes might affect the quality of the nuclei extracted using the snPATHO-seq method. Therefore, unfortunately, we could not provide a more quantitative assessment of the cells:nuclei ratio.

In addition, as mentioned above, the scFFPE has been amended to highlight its nuclei-isolating nature. Therefore, a scRNA-seq workflow for FFPE samples does not currently exist. Therefore, our study echoes previous methods, including snFFPE-seq and snRandom-seq, in pursuing nuclei isolation instead of cell isolation from FFPE samples.

The authors should list the main benefits for using snPATHO-seq over other more readily available protocols. It appears that the procedure is not easier, rather more complex than scFFPE, since there is an additional nuclear extraction step. Also, please include the composition of the nuclear extraction buffer and other details of the experimental protocol, since this is extremely beneficial for experimentalists looking to use snPATHO-seq in their work.

We thank the reviewer for this suggestion. We expanded the Results and Discussion section 4 to provide more comments on the differences between the snPATHO-seq and other workflows, including the snFFPE-seq, snRandom-seq, and scFFPE workflows.

The composition of the nuclei extraction buffer was included in the method section. A more detailed workflow was released on the protocol.io (<https://www.protocols.io/view/snpatho-seq-8epv5x58dg1b/v1>).

There were at least two more highly similar methods published recently, which should be compared to snPATHO-seq; snRandom-seq (<https://www.nature.com/articles/s41467-023-38409-5#ref-CR15>) and snFFPE-seq (<https://www.biorxiv.org/content/10.1101/2022.08.25.505257v1>). In light of this, the novelty of snPATHO-seq should be presented in the manuscript. To strengthen the conclusions of the current manuscript, kidney snPATHO-seq data could be directly compared to the

snRandom-seq results.

We thank the reviewer for this suggestion. In the Results and Discussion section 4, we compared the snPATHO-seq to these workflows. Briefly, the snFFPE-seq showed inferior performance compared to the standard single-cell workflow when tested using matching FFPE and frozen tissue samples, as the developers mentioned. On the other hand, the snPATHO-seq demonstrated comparable performance to the standard 10x 3' workflow when tested using matching FFPE and frozen samples. This is likely because the gene expression detection methods employed in the snFFPE-seq study were designed for samples with intact RNA and underperform when RNA molecules are fragmented. In addition, we conducted a direct comparison between the snPATHO-seq and the snRandom-seq using the kidney datasets. Indeed, the snRandom-seq detected many more UMIs and genes per nucleus than the snPATHO-seq workflow (Panel b). This is likely due to the variations in transcriptomic coverage between the gene expression detection chemistry employed. Nonetheless, the snPATHO-seq mainly uses off-the-shelf reagents and an established computational pipeline (developed by the 10x Genomics), making it easier for interested researchers to adapt. Diving deeper into the gene expression profiles, we identified similar cell types from snRandom-seq and snPATHO-seq kidney datasets (Panel a,c,d). The expression of canonical cell type markers also appeared to be generally concordant (Panel e).

However, we've decided to exclude these results from the manuscript because of the small sample size for this comparison. We only have 1 set of data generated by each method but have two confounding factors: species (snRandom-seq: mouse kidney; snPATHO-seq: human kidney) and technology (snRandom-seq vs snPATHO-seq). Therefore, we could not systematically investigate the differences between these technologies due to sample limitations. Adapting the snRandom-seq in our research facility is costly, as mentioned above. Therefore, instead of including these analysis results in the manuscript, we decided to focus on comparing the differences in assay design and cost-effectiveness of these two workflows.

How reproducible are cell type fractions within the same sample and between different workflows? This would be important to show in Fig.1a to show reproducibility between snPATHO-seq, Frozen-3' and Frozen-Flex. Apart from absolute numbers (presented in Extended Data Fig.2c), scaled cell fractions would allow easier comparisons between workflows.

We thank the reviewer for this comment. Scaled cell fractions have now been included to help readers evaluate the consistency of the workflows (Figure 2h, Figure S4c,g). In general, we detected a comparable fraction of cells using all workflows tested except for the 4399 data, where the liver tissue-resident cell types were only detected by the snPATHO-seq method.

Minor comments:

Fig.1c only shows the UMAP result from down-sampled data. Can the clustering results obtained with whole data be shown in the supplement?

We thank the reviewer for this comment. Down-sampling was necessary due to the variations in sequencing depth between datasets. For example, the 4066 snPATHO-seq dataset was sequenced with a total of 715,709,832 reads, corresponding to 107,739 reads per nucleus, while the 4066 Frozen-Flex dataset was sequenced with a total of 286,808,793 reads, corresponding to 35,465 reads per nucleus. It is only fair to compare these datasets after downsampling. However, down-sampling in our case didn't suggest the data complexity was compromised. The 10x Genomics recommended a minimum of 10k reads per nucleus for the Flex assay, and with a down-sampling of 25k reads per nucleus for breast cancer data, we have passed this threshold by 1.5 times.

Moreover, we have deposited the original FASTQ files with all reads generated and downsampled cellranger outputs into GEO for interested researchers to explore.

Ext Data Fig.1f&g: Instead of pairwise overlaps, the confidence scores of the label transfer would be more informative to assess the cell type annotation quality.

We thank the reviewer for this comment. The confidence scores of the label transfer have been included (Figure S2c).

Abstract could be longer and could emphasize the impact of the paper.

We thank the reviewer for this comment. The abstract has been updated accordingly.

Paper titles could be more informative and search friendly.
Thank you for this suggestion. We have provided a new manuscript title.

Version 1:

Reviewer comments:

Reviewer #1

(Remarks to the Author)

The authors have provided a comprehensive comparison of the snPATHO-seq workflow with 10x 3' and Flex methods, as well as other FFPE single-cell/nucleus RNA sequencing approaches, which significantly enhances the manuscript. In the discussion section, they noted that snRandom-seq remains a highly customized workflow requiring further commercialization. However, to my knowledge, a recently published paper introduced an automated version of snRandom-seq for single-nucleus total RNA sequencing of archival FFPE samples(Xu, et al. 2024). Additionally, M20 Genomics has commercialized snRandom-seq, releasing both reagent kits and instruments. I suggest the authors include this reference and update their discussion to reflect the current state of snRandom-seq, which would provide a more accurate and up-to-date comparison. I believe that with these revisions, the manuscript will be suitable for publication.

Reference:

Xu, Z., Chen, L., Lin, X., Lyu, Y., Zhou, M., Chen, H., Zhang, H., Zhang, T., Chen, Y., Suo, Y., Liang, Q., Qin, Z., & Wang, Y. (2024). Single Nucleus Total RNA Sequencing of Formalin-Fixed Paraffin-Embedded Gliomas. *Small methods*, e2301801. Advance online publication. <https://doi.org/10.1002/smt.202301801>

Reviewer #2

(Remarks to the Author)

I would like to thank the authors for revising the manuscript. The authors have addressed my concerns.

Reviewer #3

(Remarks to the Author)

Thank you for revising the manuscript. Most of my concerns and suggestions were properly addressed and I recommend the manuscript for publication.

I have some trouble reading the figure text, so I would suggest that font size is increased and higher resolution images are included in the final version of the manuscript.

Open Access This Peer Review File is licensed under a Creative Commons Attribution 4.0 International License, which permits use, sharing, adaptation, distribution and reproduction in any medium or format, as long as you give appropriate credit to the original author(s) and the source, provide a link to the Creative Commons license, and indicate if changes were

made.
